# The Concept of “Hypersexuality” in the Boundary between Physiological and Pathological Sexuality

**DOI:** 10.3390/ijerph20105844

**Published:** 2023-05-17

**Authors:** Giulio Perrotta

**Affiliations:** Istituto per lo Studio delle Psicoterapie—ISP, Via San Martino Della Battaglia n. 31, 00185 Rome, Italy; info@giulioperrotta.com

**Keywords:** hypersexuality, nymphomania, satyriasis, personality disorders, bipolarism, sexual arousal

## Abstract

Introduction: The concept of hypersexuality belongs to modern parlance, according to a predominantly clinical meaning, and is understood as a psychological and behavioural alteration as a result of which sexually motivated stimuli are sought in inappropriate ways and often experienced in a way that is not completely satisfactory. Methods: Literature up to February 2023 was reviewed, with 25 searches selected. Results: Forty-two articles were included in the review. Conclusion: Hypersexuality is a potentially clinically relevant condition consisting of one or more dysfunctional and pathological behaviours of one’s sexual sphere and graded according to the severity of impairment of subjective acting out; for this reason, the Perrotta Hypersexuality Global Spectrum of Gradation (PH-GSS) is suggested, which distinguishes high-functioning forms (pro-active and dynamic hypersexuality) from those of attenuated and corrupted functioning (dysfunctional and pathological hypersexuality of grades I and II). Future research is hoped to address the practical needs of this condition, such as the exact etiopathology, the role of oxytocin in dopaminergic hypotheses (and its ability to attenuate the symptomatology suffered by the patient in terms of manic drive), the best structural and functional personality framing of the subject, and the appropriate therapy to pursue.

## 1. Introduction

### 1.1. General and Historical Profiles

The concept of “hypersexuality” belongs to modern parlance, according to a predominantly clinical meaning, understood as a psychological and behavioural alteration as a result of which sexually motivated stimuli are sought in inappropriate ways, often experienced in a way that is not completely satisfactory; it is a psychopathological label strongly desired by the scientific community to replace terms previously used in other areas of study as well, such as nymphomania and satyriasis, the former referring to the female sexual gender and the latter to the male sexual gender [1]. Such terminologies draw on Greek mythological culture, referring to the nymphs, who were young and beautiful maidens whose beauty attracted the desire of many men. In fact, according to mythological tradition, it was they who first used the art of seduction to continually procure new sexual partners and thus the satisfaction of their pleasures. However, the nymphs, like the sirens, hid a dark side: their company was as deadly as it was pleasurable. They often fell prey to the satyrs, bearded beings who were half-man, half-animal (goat or horse), and also devoted to lustful wildlife. The dances of the satyrs and nymphs are depicted in very famous works, which capture precisely the peculiarity and complementarity of their sexual behaviour. Nymphomania and satyriasis are, therefore, the female and male equivalents of today’s hypersexuality, which, in the past, was considered a morbid accentuation of sexual impulses. In later centuries, popular tradition extolled these aspects in many artistic representations, thus giving rise to myths and stories, widely using the terms now replaced with hypersexuality beginning in 1771 A.D. in the edited work “Nymphomania. Treatise on Uterine Fury” by French physician Giambatist De Bienville [2].

### 1.2. Clinical Profiles

Today, hypersexuality is proposed differently and with a predominantly clinical connotation; in fact, it is present in the two main diagnostic manuals of psychiatric problems: the International Classification of Diseases (ICD) [3] and the Diagnostic and Statistical Manual of Mental Disorders (DSM-5-TR) [4]. The ICD is the international classification proposed by the World Health Organization; it is not specific to mental disorders alone, to which, however, a section is dedicated anyway, and it encapsulates the diagnostic criteria for all possible organic disorders. The DSM-5-TR, on the other hand, is the proposal from the American Psychiatric Association group and, currently, the most widely used classification in the international scientific community, being devoted entirely to psychopathologies. However, if the two classification systems are equivalent in terms of formulation and representation, this is not true for several clinical hypotheses, including hypersexuality itself; this is an expression of a difficulty on the part of the expert editors to frame this particular clinical case in a structural and sharable way. Hypersexuality falls under “Sexual Compulsive Behaviour Disorder” (code 6C72) for ICD-11, while for DSM-5 it is considered a behavioural addiction that can characterise the subject’s pathological conduct and even be a dysfunctional trait in other disorders. The diagnostic criteria advanced by the World Health Organisation on the subject of hypersexuality are: (A) for at least six months, recurrent and intense sexual fantasies, sexual urges, or sexual behaviour in association with three or more of the following: time spent in repetitive sexual fantasies, impulses, or behaviours that interfere with other important (non-sexual) goals, activities, or obligations; repetitive engagement in sexual fantasies, impulses, or behaviours in response to dysphoric mood states (e.g., anxiety, depression, boredom, or irritability); repetitive engagement in sexual fantasies, impulses, or behaviours in response to stressful life events; repetitive but unsuccessful efforts to control or significantly reduce such fantasies, impulses, or behaviours; repetitive engagement in sexual behaviours, disregarding the risk of physical or emotional harm to self or others; (B) there is clinically significant personal distress or impairment in social, work, or other important areas associated with the frequency and intensity of these sexual fantasies, impulses, or behaviours; (C) these sexual fantasies, impulses, or behaviours are not a direct result of medical conditions (e.g., brain tumours or dementia) or substance intake (e.g., a substance of abuse or medication) [3,5].

### 1.3. Aim

A review was conducted to determine as precisely as possible the exact aetiology of pathological hypersexuality in its various meanings. The object of this discussion is to try to determine whether, in the current state of scientific knowledge, the following research questions can be answered: Is it possible to distinguish one or more forms of hypersexuality? And if so, is there a pathological form, and how does it differ from the other hypotheses?Is it possible to say that pathological hypersexuality is a clinical condition that is medically relevant, or is it a subjective maladaptive behaviour? And if it is a condition, is it primary or secondary?Is it possible to identify the etiological hypotheses of pathological hypersexuality?

## 2. Materials and Methods

### 2.1. Study Design

Literature up to February 2023 was reviewed.

### 2.2. Review Questions

To identify the important aspects of the examination, the writer focused on the elements that could determine whether or not pathological hypersexuality could be considered a nosographic category in its own right or whether it could fall under obsessive-compulsive disorders, impulsive conduct disorders, or behavioural addictions, up to the hypothesis that interprets hypersexuality as one of the typical features of certain psychiatric disorders (e.g., bipolar, borderline personality disorder, and pathological narcissism).

### 2.3. Materials and Methods

The author searched PubMed until April 2023 for systematic reviews, clinical trials, and randomised controlled trials using the keyword “hypersexuality”, selecting 245 useful results. To have a greater and more complete overview of the topic, we ultimately selected a total of 25 studies, still adding 26 more reviews to be able to argue the elaborated content (to more easily contextualise definitions and clinical-diagnostic profiles), for an overall total of 51 results (Figure 1). Simple reviews, opinion contributions, or publications in popular volumes were excluded because they were not relevant or redundant for this work. The search was not limited to English-language papers (Table 1).

## 3. Results

### 3.1. The Different Forms of Hypersexuality

No work appears in the literature that clearly and precisely differentiates the different forms of hypersexuality. Although the shared use of this terminology has replaced the previous terms “nymphomania” and “satyriasis”, no one has ever put in place a real classification capable of differentiating and grading hypersexuality. This first limitation thus confuses both the structure and function of the term concerning the clinical profiles investigated here. It could be argued that the concept of hypersexuality in itself is too general and does not apply exclusively to pathological hypotheses; it is, therefore, necessary to expand the knowledge derived from the current definitions and try to derive from them what is necessary to arrive at a possible separation of terms. We know that hypersexuality becomes pathological: (a) when the subject experiences clinically significant discomfort concerning his or her sexual conduct, which, however, does not always coincide with an inner discomfort in realising one’s urges but depends on the social judgement of the community in which he or she lives (be it the couple, friendship, family, work or social nucleus); (b) when the subject manifests the presence of a significant and disproportionate increase in the sexual drive for at least 6 months, which, however, does not always correspond to the satisfaction of one’s sexual desire, and consisting of recurrent and intrusive fantasies and thoughts, excessive sexual behaviour, and an inability to control one’s sexuality, despite the negative consequences resulting from it; (c) when the subject, in the realisation of one’s sexual drives, engages in paraphiliac dysfunctional conduct, thus in comorbidity with them or with sexual dysfunction [14,28,29] (Table 2).

### 3.2. Hypersexuality as a Clinical Condition or Maladaptive Behaviour?

There is still debate in the scientific community about the age-old question of whether hypersexuality is a maladaptive subjective behaviour due to a traumatic psychophysical event or is rather a clinical condition. There is no definitive answer, and this gap prevents further exploration of the issue; however, by reviewing what is in the literature, it is possible to define hypersexuality as a psychological and behavioural alteration as a result of which people seek sexually motivated stimuli in inappropriate ways, often in a manner that is not completely satisfactory [1]. This definition indirectly allows hypersexuality to be labelled as a potentially clinically relevant condition, the result of one or more subjective behaviours that are maladaptive to the expected statistical mean. By this assumption, there is always debate in the literature as to whether hypersexuality is a primary or secondary condition—that is, whether or not it depends on other causes. Today, the majority position seems to reasonably lean towards the secondary hypothesis, as it appears to be a dysfunctional adaptation to a primary fact that secondarily induces the onset of the hypersexual condition. In fact, all etiological hypotheses underlying hypersexuality confirm this reasoning: (a) neurological syndromes [7,8,10,13,16,20,23,30,31,32,33,34], such as Klüver-Bucy syndrome (consisting of a bilateral lesion of the amygdala), typical and atypical dementias with temporo-frontal involvement, Pick’s dementia, Kleine-Levin syndrome (or recurrent hypersomnia), autism, and attention-deficit/hyperactivity disorder (ADHD); (b) psychiatric forms [9,11,12,21,24,25,27,35,36,37,38,39], such as bipolar disorder and borderline disorder (in their euphoric/sub-euphoric components typical of manic and/or hypomanic), sub-obsessive forms, sexually oriented behavioural addictions, and high-functioning personality disorders, such as covert-type narcissism; (c) traumas of the encephalon [40], in the regions used for rationality and impulse control (temporofrontal and limbic system in general); (d) implications arising from the use of excitatory drugs (such as methamphetamine, cocaine, synthetic drugs, and hallucinatory drugs) and from the therapeutic use of certain drugs, such as the use of L-dopa and prolactin inhibitors in Parkinson’s dementia (indeed, dopaminergic drugs have been shown to influence conscious processing of rewarding stimuli and are associated with impulsive-compulsive behaviours, such as hypersexuality, by going on to activate the nucleus accumbens and dorsal anterior cingulate when shown subliminal sexual stimuli), anabolic drugs, and testosterone and other sex hormone products [17]. It is precisely in the presence, therefore, of the fulfilment of generally accepted diagnostic criteria, out of any other physiological condition (albeit deviating from the statistical mean of the reference population) that the diagnosis of sexual conduct dysfunction due to hypersexuality is reached, resulting in the evaluation of a multidisciplinary therapeutic approach considering individual and/or group psychotherapy, mainly of cognitive-behavioural or constructivist-strategic approach (to correct reinforcers and coping strategies, work on one’s emotions, motivational recovery and metacognitive functions), and psychopharmacological treatment based on symptomatology, with anxiolytics, antidepressants, mood stabilisers, and antipsychotics [15,18,19,41] (Table 2).

### 3.3. The Etiopathological Theories of Hypersexuality

Several etiopathological theories in the literature try to explain hypersexuality, but all of them do not seem to be fully satisfactory enough to answer the question in a supportable and reproducible way. However, if the compulsive, impulsive, and psycho-traumatic models can partially explain the hypersexual condition, the neurobiological model manages to be more precise. Below is a schematization of the main etiopathological theories (Table 3).

## 4. Discussion and Limitations

The pathological concept of “hypersexuality” and its clinical reference are well established in the literature; however, it is not possible to clarify whether only that person who manifests such behaviours because of his promiscuity or even if he enacts them exclusively with one person, but with amplified frequency mode, should be considered “hypersexual.” In fact, at present, it is not even possible to clarify whether pathological hypersexuality depends only on the number of relationships (understood in a quantitative sense) or also on the intensity of them (understood in a qualitative sense), because of the time spent and resources used. 

Although there are no established criteria for hypersexuality, traits that are commonly seen in a hypersexual person or sexual addict include [46]:

Sex obsession. Spending a lot of time fantasising about sexual urges and enacting sexual behaviours.Compulsive and frequent masturbation (one to several times a day).Use of virtual pornography. Sources include videos, adult magazines, and the Internet (websites, webcams). Masturbation with virtual material is practised with repeated and systematic frequency.Massive dedication in a daily time frame, planning sexual activity. One spends several hours deciding where and how to get the next sexual “pleasure”.Use of sexual services. This already represents a more structured and pathologically dysfunctional behaviour, as sexual activities now involve direct or indirect human interaction. Behaviours may include phone sex, connections via Internet chat rooms, paying for sexual encounters, visits to strip clubs, multiple partners, or frequent one-night stands.Degeneration of behaviour into reckless, socially objectionable, or illicit sexual activity. Substance abuse, sexual assault, or dangerous sexual activity (such as autoerotic asphyxiation) may be added to sexual activity.Engaging in sexual behaviour that goes against one’s values, religious beliefs, or what society deems appropriate, experiencing helplessness, or inability to restrain oneself.Manifesting paraphiliac attitudes or behaviours. These are sexual behaviours that result in psychological distress, injury, or the death of another person. Examples include exhibitionism (exposing genitals to strangers), voyeurism (watching or participating in sexual activity with others), sadomasochism (sexual pleasure in inflicting pain or humiliation on others), and paedophilia (sexual feelings towards children).Inability to stop one’s sexual behaviour despite negative consequences in other areas of personal life, such as emotional, love, family, social, professional, or health-related relationships.

The physiological and pathological boundary seems to be marked exclusively by adherence or non-adherence to the shared and widely used diagnostic criteria in psychiatric practise, but such a view seems overly stringent and impractical concerning the subjective emotional universe of the person. “Hypersexual” should therefore be considered those who, in compliance with the above diagnostic criteria, present an accentuation of the individual sexual dynamic plot of a presumed neuropsychiatric matrix of dimensional type (and not purely categorical) [47,48], such as to condition one’s own and others’ lives according to sexual activity; the more it presents itself as conditioning and limiting concerning other spheres of life (family, work, social relations, affective and sentimental relationships), in reason of its qualitative and quantitative manifestation, the more hypersexuality presents dysfunctional pathological connotations, deserving of clinical intervention. 

Another issue that is not unanimously answered in the literature concerns the relationship between hypersexuality and paraphilia. It is unclear whether hypersexuality can be considered paraphilia or is simply the summation of behaviours aimed at intensifying sexual activity beyond the limits predetermined by the reference context. In this regard, the inference that seems most consistent with the topic analysed is to consider hypersexuality as a behavioural status with a strong emotional impact that can manifest itself with or without the presence of one or more paraphilias; in essence, paraphilias become the means used by the hypersexual to fuel his dysfunctional state, but they may not be present, may be limited to a precise context, or still be attenuated or nuanced concerning the main conduct of seeking sexual activity. A hypersexual who has one or more paraphilias structured into a paraphiliac disorder is markedly pathological, compared to a hypersexual who does not have such a disorder in his or her personological profile.

Yet another issue concerns the relationship between hypersexuality and mental disorders as nosographically framed in psychiatry. The literature here is rich on this topic, and hypersexuality is always associated with pathological and dysfunctional personality frameworks, as is the case with borderline, histrionic, narcissistic, and antisocial disorders, but also with markedly manic bipolar profiles, in obsessive-compulsive disorder, sadistic and masochistic personalities, and in psychosis with erotically motivated features; however, it is not clear whether hypersexuality is the factor that fuels dysfunctionality in these morbidities or is rather a constitutive element of them [49,50].

No research in the literature has focused on the possible role of oxytocin in hypersexuality, and thus the neurobiological relationship with dopaminergic accesses is instead widely studied in sexual conduct-related disorders; however, some authors have hypothesised this correlation on the basis of oxytocin effects on brain areas involved in emotional processes [51,52].

Various psychometric instruments are used to assess the degree of functional impairment of hypersexual individuals, such as in the case of the Sex-relation Evaluation Schedule Assessment Monitoring (SESAMO), which explores sexual and relational aspects, the Sexual Addiction Screening Test (SAST) and the Sexual Compulsivity Scale (SCS), which investigate the compulsive component of dependent sexuality, of the Hypersexual Behaviour Inventory (HBI), which investigates hypersexual behaviours, and the Childhood trauma questionnaire (CTQ), which investigates psycho-traumatic profiles of the infant; however, none of these instruments clearly and comprehensively define a graded scale of hypersexuality capable of distinguishing between highly functional and dysfunctional forms. 

Because of these limitations, it was decided to propose a scale to comprehensively assess the severity of the symptoms of hypersexuality, capable of collecting the range of the main hypotheses and thus grading the behaviour according to the different factors involved. This classification (*Perrotta Hypersexuality Global Spectrum of Gradation*, **PH-GSS**) therefore intends to fill the content gap found here in the literature and propose a comprehensive measure, pending validation using a representative sample. The following table shows the content of the PH-GS (Table 4).

## 5. Conclusions

Hypersexuality is the term that describes a series of heterogeneous conditions, substantiated by behaviours of a sexual nature, that must be graded according to their severity of impairment of subjective agency. If the terms nymphomania and satyriasis now appear judgmental, it is only correct to speak of hypersexuality if a grading scale is put in place to help the practitioner distinguish between the different hypotheses. For this reason, the *Perrotta Hypersexuality Global Spectrum of Gradation* (**PH-GSS**) is suggested, which distinguishes high-functioning forms (pro-active and dynamic hypersexuality) from those of attenuated and corrupt functioning (dysfunctional and pathological grade I and II hypersexuality). Hypersexuality, therefore, is a potentially clinically relevant condition secondary to another medical condition and consisting of one or more dysfunctional and pathological behaviours of one’s sexual sphere, which can be explained in a more shareable and reproducible way according to the neurobiological theoretical model. Future research is hoped to address the practical needs of this condition, such as the exact etiopathology, the role of oxytocin in the dopaminergic hypotheses (and its ability to attenuate the symptomatology suffered by the patient in terms of manic drive), the best structural and functional personality framing of the subject, and the appropriate therapy to be pursued.

## 6. Implications for Clinical Practise

Systematically defining the gradation of hypersexuality means being able to frame the subject more structurally and functionally by choosing the most appropriate treatment and avoiding letting the therapist decide based on his or her subjective assumption or interpretation of the patient’s symptomatic experience. It is therefore extremely important to be able to complete such an operation to give clinical dignity to a condition that, to date, has not been fully explained and is often interpreted only as a personality trait or accentuation of the sexual pattern because of the primary medical condition.

## Figures and Tables

**Figure 1 ijerph-20-05844-f001:**
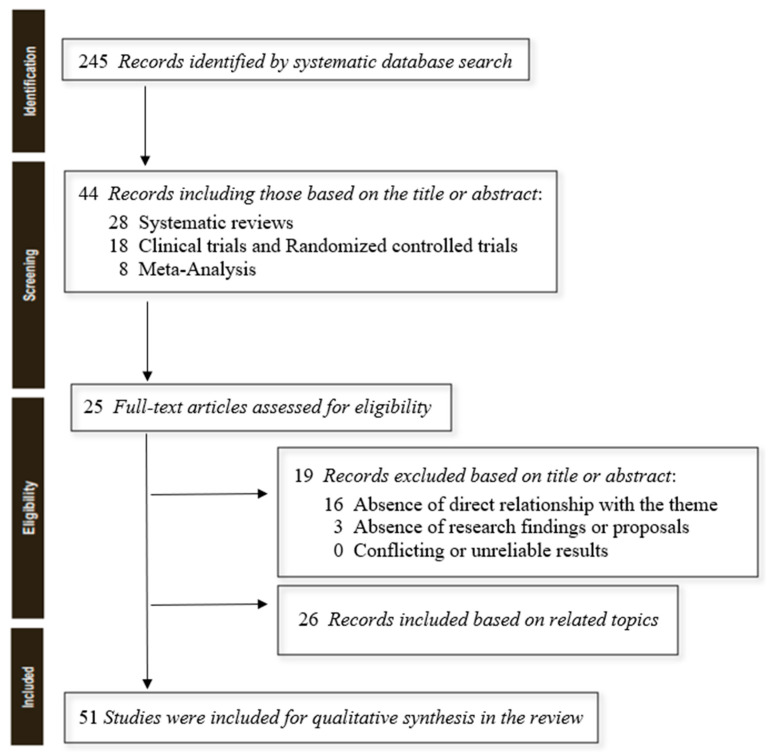
PRISMA flow diagram template for systematic reviews. Adapted from Matthew J. Page et al., BMJ 2021; 372:n71 [6].

**Table 1 ijerph-20-05844-t001:** Selected manuscripts on the theme of hypersexuality. RES (clinical trial or randomised controlled trial); M (meta-analysis); R (review and systematic review); E (editorial).

Author (Year)	Objectives	Type	Key Results and Conclusions
Korchia, T. et al. (2022) [7]	ADHD prevalence in patients with hypersexuality and paraphilic disorders	M	Attention deficit hyperactivity disorder (ADHD) is much more frequent in populations with hypersexuality or paraphilic disorders compared to the general population.
Burley, C.V. et al. (2022) [8]	Pharmacological and nonpharmacological approaches and dementia	R	Pharmacological and nonpharmacological approaches to reduce disinhibited behaviours in dementia.
Landgren, V. (2022) [9]	Effects of testosterone suppression on desire, hypersexuality, and sexual interest in children in men with pedophilic disorder	RES	The effects of testosterone withdrawal on significant correlates of paedophilic disorder (PeD) are largely unknown. The purpose of this study was to explore in detail the effects of testosterone suppression from degarelix as compared to placebo on desire, hypersexuality, and subjectively experienced sexual interest in participants with PeD. We compared the sexual effects of degarelix, a GnRH antagonist, on men with PeD assigned to degarelix or placebo. Sexual Desire Inventory scores decreased significantly at two weeks and ten weeks in participants assigned degarelix, whereas HBI ratings did not differ significantly at two weeks but did so at ten weeks. Fifteen out of twenty-six individuals (58%) in the group given degarelix and three out of twenty-six (12%) in the group given the placebo reported no further sexual interest in children at ten weeks.
Latella, D. et al. (2021) [10]	Hypersexuality in neurological diseases	M	Hypersexuality is a frequent sexual disorder in patients with neurological disorders, especially neurodegenerative ones.
Jennings, T.L. (2021) [11]	Compulsive sexual behaviour, religiosity, and spirituality	R	Although research examining compulsive sexual behaviour (CSB) and religiosity has flourished, such growth is hampered by cross-sectional samples lacking in diversity. Moral incongruence assists in explaining the relationship between religiosity and spirituality, but future research should consider other manifestations of CSB beyond spirituality.
de Oliveira, L. et al. (2020) [12]	The link between boredom and hypersexuality	R	Although current literature identifies a link between boredom and hypersexuality, further substantive research is still needed to clarify the associations between the two constructs.
Soldati, L. et al. (2020) [13]	Sexual function/dysfunctions and ADHD	M	Attention deficit hyperactivity disorder (ADHD) is a mental disorder affecting sexual health. Further studies are warranted to learn more about sexuality in subjects with ADHD.
Parra-Dìaz, P. et al. (2020) [14]	Impulse control disorders (ICDs) and Parkinson’s disease (PD)	R	The tendency towards a different ICD profile in different geographical areas may be attributable to socio-economical, cultural, or political influences in the phenomenology of these disorders.
Schecklmann, M. (2020) [15]	Repetitive transcranial magnetic stimulation as a potential Tool to reduce sexual arousal	RES	The results indicate that 1 session of high-frequency rTMS (10 Hz) of the right dorsolateral prefrontal cortex (DLPFC) could significantly reduce subjective sexual arousal induced by visual stimuli in healthy subjects.
De Alarcon, R. et al. (2019) [16]	Alarcòn Online porn addiction	R	Hypersexual disorder fits this model and may be composed of several sexual behaviours, like problematic use of online pornography (POPU).
Montgomery-Graham, S. (2017) [5]	Conceptualization and assessment of hypersexual disorder	R	The Hypersexual Disorder Screening Inventory, the measure proposed for the clinical screening of hypersex-disorder (HD) by the Diagnostic and statistical manual of mental disorders (DSM) workgroup, currently has the strongest psychometric support.
Walton, M.T. (2017) [17]	Hypersexuality	R	Our discussion of hypersexuality covers a diversity of research and clinical perspectives.
Hallberg, J. (2017) [18]	A cognitive-behavioural Therapy Group Intervention for Hypersexual Disorder	RES	Although participants reported decreased hypersex-disorder (HD) symptoms after attending the cognitive-behavioural therapy (CBT) programme, future studies should evaluate the treatment programme with a larger sample and a randomised controlled procedure to ensure treatment effectiveness.
Hallberg, J. (2017) [19]	A randomised controlled study of group-administered cognitive behavioural therapy for hypersexual disorder in men	RES	A significantly greater decrease in hypersex-disorder (HD) symptoms and sexual compulsivity, as well as significantly greater improvements in psychiatric well-being, were found for the treatment condition compared with the waitlist. These effects remained stable at 3 and 6 months after treatment.
Nakum, S., Cavanna, A.E. (2016) [20]	Hypersexuality in Parkinson’s disease	R	Hypersexuality is not rare in patients with Parkinson’s disease (PD) treated with brain dopamine replacement therapy (DRT), particularly in those on dopamine agonists.
Castellini, G. et al. (2016) [21]	Sexuality in eating disorder patients	R	The analysis of the literature showed an association between sexual orientation and gender dysphoria with eating disorders (ED) psychopathology and pathological eating behaviours, confirming the validity of research developing new models of maintaining factors of EDs related to the topic of self-identity.
Chatzittofis, A. (2016) [22]	HPA-axis dysregulation in men with hypersexual disorder	RES	Hypersexual disorder, integrating pathophysiological aspects such as sexual desire deregulation, sexual addiction, impulsivity, and compulsivity, was suggested as a diagnosis for the DSM-5. The patients reported significantly more childhood trauma and depression symptoms compared to healthy volunteers. The diagnosis of hypersexual disorder was significantly associated with Dexamethasone suppression tests (DST) non-suppression and higher plasma DST-ACTH (Adrenocorticotropic Hormone), even when adjusted for childhood trauma. The results suggest Hypothalamic–pituitary–adrenal (HPA) axis dysregulation in male patients with hypersexual disorder.
de Oliveira, M. et al. (2016) [7]	Pharmacological treatment for Kleine-Levin syndrome	R	Therapeutic trials of pharmacological treatment for Kleine-Levin syndrome with a double-blind, placebo-controlled design are needed.
Codling, D. et al. (2015) [23]	Hypersexuality in Parkinson’s disease	R	A brief survey of the neurobiology of sexuality suggests possible avenues for further research and treatment of hypersexuality (HS).
Karila, L. et al. (2014) [24]	Sexual addiction or hypersexual disorder	R	Addictive, somatic, and psychiatric disorders coexist with sexual addiction.
Schultz, K. et al. (2014) [25]	Nonparaphilic hypersexual behaviour and depressive symptoms	M	There was a moderately positive relationship between non-paraphilic hypersexual behaviour and depressive symptoms. This relationship was similar across gender, sexual orientation, and age.
Oei, N.Y.L. et al. (2012) [26]	Dopamine modulates reward system activity during the subconscious processing of sexual stimuli	RES	Brain activation was assessed during a backwards-masking task with subliminally presented sexual stimuli. Results showed that levodopa significantly enhanced the activation in the nucleus accumbens and dorsal anterior cingulate when subliminal sexual stimuli were shown, whereas haloperidol decreased activations in those areas. Dopamine thus enhances activations in regions thought to regulate ‘wanting’ in response to potentially rewarding sexual stimuli that are not consciously perceived. This running start of the reward system might explain the pull of rewards in individuals with compulsive reward-seeking behaviours such as hypersexuality and patients who receive dopaminergic medication.
Kowatch, R.A. et al. (2005) [27]	Phenomenology and clinical characteristics of mania in children and adolescents	M	The clinical picture that emerges is that of children or adolescents with periods of increased energy (mania or hypomania), accompanied by distractibility, pressured speech, irritability, grandiosity, racing thoughts, decreased need for sleep, and euphoria/elation.

**Table 2 ijerph-20-05844-t002:** Research questions/answers.

Questions	Answers
Is it possible to distinguish one or more forms of hypersexuality? And if so, is there a pathological form, and how does it differ from the other hypotheses?	Currently, it is not possible in the literature to distinguish between the different forms of hypersexuality except by trying to identify the form deemed pathological because it meets the diagnostic criteria in psychiatry. For this reason, the Perrotta Hypersexuality Global Spectrum of Gradation (PH-GSS) is suggested and recommended.
Is it possible to say that pathological hypersexuality is a clinical condition that is medically relevant, or is it a subjective maladaptive behaviour? And if it is a condition, is it primary or secondary?	Hypersexuality is a potentially clinically relevant condition secondary to another medical condition (encephalic trauma, neurological, drug, substance abuse, or psychiatric) and consisting of one or more dysfunctional and pathological behaviours in one’s sexual sphere.
Is it possible to identify the aetiology of pathological hypersexuality with scientific certainty, thus in a reproducible and agreeable manner?	Several etiopathological theories in the literature try to explain hypersexuality, but all of them do not seem to be fully satisfactory enough to answer the question in a supportable and reproducible way. However, if the compulsive, impulsive, and psycho-traumatic models can partially explain the hypersexual condition, the neurobiological model manages to be more precise.

**Table 3 ijerph-20-05844-t003:** Table of major etiopathological theories of hypersexuality.

Theoretical Models	Content
*Compulsive*	Hypersexuality is a modality of obsessive-compulsive disorder (OCD), in that it manifests as recurrent and intense sexual fantasies that interfere with the performance of normal daily activities, while compulsions could be configured as sexual behaviours that are very difficult to counteract and take up a lot of the person’s available time. In addition, the sexual act (whether masturbatory or through intercourse with a partner) is seen as a libidinal drive release mechanism that increases during times of high stress. However, unlike OCD, thoughts and behaviours related to sexuality are egosyntonic, i.e., consistent with one’s self; they do not create discomfort for the person and are seen as natural and devoid of any problematic aspects, on par with personality disorders. Those suffering from obsessive-compulsive disorder, on the other hand, perceive obsessions as highly intrusive. Hypersexuality cannot, therefore, be part of the category of obsessive-compulsive and related disorders, strictly speaking.
*Impulsive*	The impulsive dysfunctional model is the one deemed correct by the World Health Organization, in which hypersexuality is seen as an impulse control disorder. Underlying it is the idea that the hypersexual person is unable to adequately manage his or her sexual impulses; he or she would act on them without modulating them the moment he or she felt them. This presupposes a sexual tension that cannot be procrastinated before the act and its release during the act, which would be followed by guilt. The lack of inhibition is due to a frontal lobe malfunction. This position, however, clashes with the behavioural representation of the hypersexual subject, who is anything but impulsive in organisational acting out: although the impulse may be correctly labelled as irrepressible, in most cases the subject then comes to plan his or her activities in a lucid, rational, and methodical manner.
*Additional (neurobiology)*	The addiction model attempts to explain hypersexuality as a behavioural addiction because of the peculiar characteristics common to addictions precisely: the tendency to tolerate sexual activity (sexual intercourse is less and less satisfying); the occurrence of withdrawal symptoms in the absence of sexual activity, such as rumination, anxiety and guilt; the difficulty in reducing, or otherwise controlling, sexual behaviours (such as compulsive masturbation, heavy use of pornography, seeking sexual stimulation through the internet and social networks, cybersex practises, promiscuous, multiple and/or casual sexuality, disinterest in the risk of contracting diseases through unprotected sexual conduct, prostitution, need for infidelity); the use of more and more considerable time aimed at seeking partners; the reduction of time devoted to other activities (sociality disappears in favour of sexual activity); the act is perpetrated despite the fact that it entails negative consequences more or less impacting the subject’s personal and social life. Moreover, this model finds particular reinforcement from scientific evidence, which has been derived from studies of the neurophysiological correlates of hypersexuality. Neuroimaging techniques have revealed dysfunction in the dopaminergic and serotonergic systems, frameworks typical of addictions, precisely underlying the compulsive and uncontrolled pursuit of sexual satisfaction. The dopamine neurotransmitter emitted by neurons located in the limbic system (nucleus accumbens) would be released in a dysregulated manner in individuals with hypersexuality due to an exaggerated and disproportionate overactivity of the mesolimbic-dopaminergic and nigrostriatal pathways. In individuals with impulse dysregulation disorders and obsessive-compulsive disorder, it is precisely this function that would be affected. Although not yet definitively validated by significant scientific research, scholars have also theorised the involvement in the aetiology of hypersexuality of the neurotransmitter serotonergic, a neuronal hormone that makes people experience feelings of happiness, satiety, and fulfilment. The same reasoning must also be applied to oxytocin [42], which is directly involved in social and affective relationships and for which there are still no in-depth studies that may or may not explain the role of oxytocin in hypersexuality [43,44].
*Psychodramatic*	It distinguishes the disorder of hypersexuality from normal intense sexual activity, citing traumatic reasons behind the establishment of dysfunctional behaviour. Psychometric instruments are used to label the clinical response so that we can fully assess the patient’s functional impairment with regard to his or her sexual conduct [22,26,45].

**Table 4 ijerph-20-05844-t004:** Perrotta Hypersexuality Scale, PHS-1.

High-Functioning Pathological			
*Level*	*Colour*	*Definition*	*Behaviour*
1	Pink	Pro-active hypersexuality	The subject presents an accentuation of the sexual storyline in terms of drive needs that are higher than the statistical average of the reference population but still fall within the physiological framework or subjective normality because they do not respond to any existing nosographic pathological profile. He or she can moderate his or her behaviour and adapt to the social context, despite feeling a reasonably more significant drive present than expected due to age and individual and collective relational context. The fulfilment of these needs is embodied in a greater drive to seek and achieve them, but no dysfunctional conduct, relevant paraphiliac comorbidities, or excessively impulsive acts are present.
2	Yellow	Dynamic hypersexuality	The subject presents a marked accentuation of the sexual plot in terms of drive needs higher than the statistical average of the reference population but still falling within the physiological framework or subjective normality because they do not respond to any existing nosographic pathological profile. He is still able to moderate his behaviour and adapt to the social context, despite feeling an unreasonably greater drive than expected due to his age and individual and collective relational context. The fulfilment of these needs is embodied in a greater propulsive drive in the pursuit and realisation of these needs in practise, but dysfunctional behaviours and paraphiliac comorbidities of mild significance are already present in the absence, however, of excessively impulsive or egregious acts.
**Pathological Attenuated Functioning**			
*Level*	*Colour*	*Definition*	*Behaviour*
3	Orange	Dysfunctional hypersexuality	The subject presents a significantly marked accentuation of the sexual plot in terms of drive needs that are higher than the statistical average of the reference population and no longer within the physiological or subjective normal framework. He moderates his behaviour with difficulty, and his functional adaptation to the social context appears coarse and irreverent, precisely because of the markedly more significant sexual drive than expected, due to age and individual and collective relational context. The fulfilment of these needs is substantiated by an excessive propulsive drive in the concrete pursuit and realisation, dysfunctional conduct, paraphiliac comorbidities of moderate significance, and impulsive acts out of context.
4	Red	Pathological hypersexuality (grade I)	The subject presents a disproportionate accentuation of the sexual plot in terms of drive needs that are higher than the statistical average of the reference population and no longer within the physiological framework or subjective normality. He is hardly able to moderate his behaviour, and his functional adaptation to the social context appears out of context and often excessive, precisely because of the significantly more pronounced sexual drive than expected due to age and individual and collective relational context. The fulfilment of these needs is substantiated by an extreme drive in the concrete pursuit and realisation, and dysfunctional behaviours, paraphiliac comorbidities of serious relevance, and impulsive out-of-context acts are present; however, he is aware of his acts and is concerned about the possible negative implications in the social context (ego-dystonia) but is not fully attuned to his emotional plan.
**Pathological to Corrupt Functioning**			
*Level*	*Colour*	*Definition*	*Behaviour*
5	Purple	Pathological hypersexuality (grade II)	The subject presents a disproportionate and unreasonable accentuation of the sexual plot in terms of drive needs well above the statistical average of the reference population and no longer within the physiological or subjective normalcy framework. He is unable to moderate his behaviour, and his functional adaptation to the social context appears severely compromised, precisely because of the significantly more pronounced sexual drive than expected due to age and individual and collective relational context. He exposes himself to danger for himself and others, gives no weight to the negative consequences of his behaviour, and adopts insane, promiscuous, impulsive, and instinctive conduct. The fulfilment of such needs is substantiated by an uncontrollable propulsive drive in concrete pursuit and fulfilment, severe dysfunctional conduct, paraphiliac comorbidities of extreme pathological relevance, and impulsive and irrational acts out of context. He is not aware of his acting out (ego-syntony) and his emotional plane, although he may display emotions and feelings that seemingly prove otherwise.

## Data Availability

Not applicable.

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
