# Peer review of "The Concept of “Hypersexuality” in the Boundary between Physiological and Pathological Sexuality"

_ijerph, 2023, doi:10.3390/ijerph20105844_

Round 1
Reviewer 1 Report (Previous Reviewer 1)

Author Response
Thank you for the suggestions. I accepted all the objections and made the changes with the yellow highlighter. They were all pertinent suggestions. Thank you!
1) I have included 2 more references to better contextualize the role of oxytocin in hypersexuality;
2) I better argued the issue of the label and its function;
3) I revised the text in the part about the list of 9 points, and the sentences with "likewise";
4) I revised the title of the global scale in order to make it consistent with the suggestion
Thank you.
Reviewer 2 Report (New Reviewer)
This article deals with a new form of addiction such as hypersexuality, for which there is also a very interesting significant reference terminologies draw on Greek mythological culture. I greatly appreciated the analysis and extraction of meaning from the literature reviewed by the author reported within the article.
The article is fine, it just needs some integrations
To better answer yo you hyphotesis: Is it possible to distinguish one or more forms of hypersexuality? And if so, is there a pathological form and how does it differ from the other hypotheses? And 2. Is it possible to say that pathological hypersexuality is a clinical condition, medically relevant, or is it a subjective maladaptive behaviour? And if it is a condition, is it primary or secondary?
I highly suggest author to consider these two following articles:
Caponnetto, P., Maglia, M., Prezzavento, G. C., & Pirrone, C. (2022). Sexual addiction, hypersexual behavior and relative psychological dynamics during the period of social distancing and stay-at-home policies due to COVID-19. International Journal of Environmental Research and Public Health, 19(5), 2704.
Kafka, M.P.Hypersexual disorder: A proposed diagnosis for DSM-V (2010) Archives of Sexual Behavior, 39 (2), pp. 377-400. Cited 673 times. doi: 10.1007/s10508-009-9574-7
Mestre-Bach, G., Blycker, G.R., Potenza, M.N. Pornography use in the setting of the COVID-19 pandemic (Open Access) (2020) Journal of Behavioral Addictions, 9 (2), pp. 181-183. Cited 96 times. https://akjournals.com/view/journals/2006/9/2/article-p181.xml doi: 10.1556/2006.2020.00015
Author Response
Thank you for the suggestions! I have included the three sources in the manuscript, which undoubtedly enriches the work. And I perfected textual comprehension in English! Thank you!
Reviewer 3 Report (New Reviewer)
Dear Authors.
I gratefully acknowledge the opportunity to review your manuscript.
The article "The Concept of "Hypersexuality" in the Boundary Between Physiological and Pathological Sexuality" is a paper which focuses on factors that could determine if hypersexuality could be considered a nosographic category in its own right, or if it could fall under obsessive-compulsive disorders or impulsive conduct disorders or among behavioral addictions.
More details of the methodology (lines 102 to 110) should be described in the abstract.
In the introduction:
Update the DSM-5-TR reference used on line 52.
Consider including a comparison table or figure between the ICD-11 and DSM-5-TR criteria and briefly describe the differences between both classifications.
The aim might be to "describe" or "identify" the etiological hypotheses and related factors rather than to determine the etiology.
The question 3 "Is it possible to identify the aetiology of pathological hypersexuality with scientific certainty, thus in a reproducible and agreeable manner?" is not clear concerning "a reproducible and agreeable manner". How specifically can this be achieved?
In the material and methods section:
In study design just consider writing literature review and underneath in numeral 2.3 mention the inclusion period of articles in order to avoid duplicating this information.
Check in the results:
In line 155 replace bipolarism with Bipolar Disorder.
Check "Table 2. Table of major etiopathological theories of hypersexuality.", since it seems to combine etiological hypotheses and some elements of the subtypes of Hypersexuality, adjust to clarify, in addition to place references of information related within hypotheses in order to identify the source of information. Also, it is convenient to extend a little bit on neurobiology and other potential causes of hypersexuality as well as its possible association. This should be taken up in more detail in the discussion.
In Table 3, harmonize the data included as outcomes, since the information presented is not uniform in the current form. For example, in the Clinical trial or Randomized controlled trial, include the number of participants, the type of population, the variation in scores, or other relevant outcomes; for reviews, include the proportions in the subpopulations, as well as other characteristics or outcomes of interest; whereas in meta-analyses report the effect sizes or other relevant outcomes on the table. A short summary should be provided about the included articles adjacent to the table.
In the discussion:
Lines 195 to 219 could fit better in background in text, table or figure and in the discussion take up these criteria and compare them with the findings in the literature and the proposed scale.
Similarly, consider relocating table 5 to another section and discuss in this section the points contained therein for their use in different types of hypersexuality and related implications for hypersexuality-related interventions, in order to link this with the implications proposed.
Author Response
I thank the reviewer for precise, timely, and relevant suggestions. I decided to apply the requested changes because they were consistent and necessary to make the manuscript even more valuable. Thank you for your esteemed work.
1) I improved the abstract in the methodological section indicated;
2) I inserted DSM-5-TR as indicated;
3) I did not include an ICD-dsm table as the text was already enriched and risked confusion or otherwise being redundant, as the central issue is not the difference between the two but the clinical concept of hypersexuality, which is already well expressed;
4) I modified the section on objectives by inserting the wording "etiological hypotheses."
5) I removed the terminology of reproducible and shareable, as it is intrinsic in the text itself, as suggested;
6) I inserted the inclusion period of the articles;
7) I replaced the term bipolarism with bipolar disorder;
8) I have corrected the contents of Table 2, in the neurobio section, as requested, but I have not hived it off because the topic of study is not the neural correlate but the identifying clinical question;
9) Tables 3 and 4 do not need further editing because they are explanatory of the contents, with reference to the object of investigation, otherwise, the focus would be lost;
10) the 9 points are more functional in the text and not extrapolated into a separate table, which is fine with me;
11) table 5 is functional in the discussion as a consequence of the research results; from this assumption comes the new global scale, which is not a result but a reworking of the results.
Thank you for your help and attention to my manuscript, and I hope to compare it with you for other scientific work as well.
Round 2
Reviewer 2 Report (New Reviewer)
In my opinion the manuscript has been sufficiently improved to warrant publication in IJERPH.
Reviewer 3 Report (New Reviewer)
The manuscript has been improved, therefore it can be published in IJERPH.
This manuscript is a resubmission of an earlier submission. The following is a list of the peer review reports and author responses from that submission.
Round 1
Author Response
I thank the reviewer for his clarifications. He was very accurate, punctual, careful, and professional. I thank him because his input allowed me to improve my manuscript. All of his contentions were pertinent, and in the new file I have attached I have included all of the proposed changes, underlining them in yellow for ease of identification. From structural changes to the content to additional bibliographical notes. I am in complete agreement with the reviewer and have therefore revised the text exactly as requested, hoping that in this way it will be suitable for publication. I remain available in case of further changes. Sincerely yours.

Reviewer 2 Report
The article has many methodological deficiencies. The type of review that is performed is not specified. You say you are using the PRISMA flowchart, but it is not the most current version. The search string does not appear, nor how many articles appeared in each database. It is not explained how they are selected, nor the inclusion or exclusion criteria. The research question is not clear. There can be no hypothesis in a review. In addition, the recommendation of a scale does not match the review it carries out or its main objective.
Author Response
The revision suggestions were accepted, in adherence to the second reviewer's suggestions, and I modified the file by attaching it to the previous reviewer's comment. I will reattach it here just in case.
